# Selective Activity of an Anthocyanin-Rich, Purified Blueberry Extract upon Pathogenic and Probiotic Bacteria

**DOI:** 10.3390/foods12040734

**Published:** 2023-02-08

**Authors:** Sara Silva, Eduardo M. Costa, Manuela Machado, Rui M. Morais, Conceição Calhau, Manuela Pintado

**Affiliations:** 1CBQF Centro de Biotecnologia e Química Fina-Laboratório Associado, Escola Superior de Biotecnologia, Universidade Católica Portuguesa, Rua Diogo Botelho 1327, 4169-005 Porto, Portugal; 2Nutrição e Metabolismo, NOVA Medical School, Universidade Nova de Lisboa, Campo dos Mártires da Pátria, 130, 1169-056 Lisboa, Portugal; 3CINTESIS, Centro de Investigação em Tecnologias e Serviços de Saúde, Universidade do Porto, 4200-450 Porto, Portugal

**Keywords:** probiotic, pathogen, antimicrobial activity, blueberry extract, organic acids, short-chain fatty acids

## Abstract

Blueberry extracts have been widely recognized as possessing antimicrobial activity against several potential pathogens. However, the contextualization of the interaction of these extracts with beneficial bacteria (i.e., probiotics), particularly when considering the food applications of these products, may be of importance, not only because their presence is important in the regular gut microbiota, but also because they are important constituents of regular and functional foodstuffs. Therefore, the present work first sought to demonstrate the inhibitory effect of a blueberry extract upon four potential food pathogens and, after identifying the active concentrations, evaluated their impact upon the growth and metabolic activity (organic acid production and sugar consumption) of five potential probiotic microorganisms. Results showed that the extract, at a concentration that inhibited *L. monocytogenes*, *B. cereus*, *E. coli* and *S. enteritidis* (1000 μg mL^−1^), had no inhibitory effect on the growth of the potential probiotic stains used. However, the results demonstrated, for the first time, that the extract had a significant impact on the metabolic activity of all probiotic strains, resulting in higher amounts of organic acid production (acetic, citric and lactic acids) and an earlier production of propionic acid.

## 1. Introduction

Blueberries are recognized as being rich in phenolic compounds, particularly anthocyanins. This trait has made them the focus of several studies, which aim to better understand the health-promoting properties of blueberry-based extracts. One of the most common attributes associated with phenolic compounds is their potential as antimicrobial agents, with several authors having demonstrated it [1,2,3,4,5,6]. However, in a previous work it has been reported that a blueberry extract, while capable of inhibiting the growth of several potential pathogens, had no inhibitory effect upon *Lactobacillus rhamnosus*, *Lactococcus lactis* and *Lactobacillus bulgaricus* growth [7]. 

This has raised an interesting question—if a blueberry extract is capable of inhibiting a pathogen, while simultaneously not inhibiting potentially probiotic microorganisms, can it be used as a potential antimicrobial additive for fermented foods or as a coadjuvant for the treatment of intestinal infections? This hypothesis is further substantiated as the body of literature supports that phenolic compounds may have a dual effect upon the gut microbiota ecosystem by exhorting both inhibitory and growth-promoting stresses upon it, with authors suggesting the term ‘duplibiotic’ to characterize this effect [8,9,10,11].

Furthermore, although a few works may be found on anthocyanin-rich extracts’ lack of inhibitory effect against potential probiotics, to the best of our knowledge none has considered the impact that the extracts’ presence may have upon their metabolic activity, when present at the concentrations needed to have an antimicrobial effect upon pathogenic microorganisms [4,7,12,13,14]. Therefore, the present work aimed to assess the impact of an adsorbent resin-purified, anthocyanin-rich, blueberry extract upon probiotic growth and metabolic activity (in particular production of organic acids) at a concentration in which the extract is effectively capable of inhibiting the growth of four potential food pathogens (*B. cereus*, *S. enteritidis*, *L. monocytogenes* and *E. coli*). This was made in an attempt to demonstrate a double potential of said extract by exerting a positive stimuli upon potentially beneficial microorganisms while effective in managing non-beneficial ones.

## 2. Materials and Methods

### 2.1. Extract Production and Purification

The extracts were produced using an ethanolic solid–liquid extraction and purified using solid phase extraction columns (Bond Elut Plexa, Agilent Technologies, Santa Clara, CA, USA) as described elsewhere [11]. The resulting powder, henceforth referred to as the extract, contained 637 mg g^−1^ of anthocyanins with all 15 different anthocyanins found in blueberries (malvidin, cyanidin, delphinidin, petunidin and peonidin arabinosides, glucosides and galactosides) being present in the extract.

### 2.2. Microorganisms

Four potential food pathogens and five different probiotic strains were considered in the present work. The probiotics considered were *Lactobacillus acidophilus* Ki, *L. plantarum* 299V, *L. rhamnosus* R11, *Bifidobacterium animalis* Bb12 (B. Bb12) and *B. animalis* Bo (B. BO) and the pathogens were *Escherichia coli* NCTC 9001, *Salmonella enteritidis* ATCC 13076, *Listeria monocytogenes* ESB 3562 (a food isolate from Escola Superior de Biotecnologia’s culture collection, Porto, Portugal) and *Bacillus cereus* NCTC 2599.

### 2.3. The Effect on Pathogenic Bacteria

#### 2.3.1. Time-Inhibition Curves

Extracts at 1000, 500, 250 and 125 μg mL^−1^ were prepared using Tryptone Soy Broth (TSB, Biokar Diagnostics, Beauvais, France), sterilized using a 0.22 μm filter (Millipore, MA, USA) and inoculated at 1% (*v*/*v*) using an overnight inoculum (ca. 108 CFU mL^−1^). The mixtures were incubated in a 96-well microplate (Nunc, Darmstadt, Germany) for 24 h at 37 °C, with the optical density (OD) at 660 nm being assessed at 1 h intervals (Fluorostar Optima Microplate Reader, BMG Labtech, Ortenberg, Germany). A positive control was drawn using inoculated culture media and sterile TSB was used as a negative control. Each condition was assayed in three independent assays, each considering triplicate of analysis [1,15].

#### 2.3.2. The Impact on Pathogenic Viable Counts

A 1000 μg mL^−1^ extract solution in TSB was inoculated with an overnight inoculum of each of the pathogenic microorganisms and incubated at 37 °C for 24 h. At the 0, 6, 12 and 24 h mark, the total viable cells were determined using decimal dilutions and plated in Plate Count Agar (PCA, Biokar Diagnostics, Beauvais, France) [16,17]. The PCA plates were then incubated at 37 °C for 24 h. Each condition was assessed in three independent assays, each considering triplicate of analysis and plating in duplicate.

### 2.4. The Effect on Probiotic Bacteria

The effect of a 1000 μg mL^−1^ extract solution prepared using de Mann Rogosa and Sharpe Broth (MRS broth, Biokar Diagnostics, Beauvais, France) for lactobacilli or MRS broth supplemented with 0.5 g L^−1^ L-cysteine-HCl (Sigma, St. Louis, MO, USA) (MRS + CYS broth) for bifidobacteria. This mixture was inoculated using an overnight inoculum and incubated at 37 °C for 24 h (bifidobacteria were incubated in anaerobiosis). At 0, 6, 12 and 24 h, the total viable cells, environmental pH values and organic acid production/sugar consumption were assessed. The total viable probiotic counts were determined using decimal dilutions and plated in either MRS (48 h at 37 °C) or MRS + CYS (48 h at 37 °C under anaerobic conditions) agar. The culture media pH values were measured using a Crison micropH 2002 (Crison Instruments S. A., Barcelona, Spain) pH reader. Sugar consumption/organic acid production was evaluated using an HPLC-RI-UV system, following the analytic conditions described by Sousa et al. [18]. Positive controls were drawn through inoculation of the respective culture media without extract and non-inoculated culture media (with and without extract) was used as a negative control. Each condition was assessed in considering three independent assays, each considering triplicate of incubations and duplicate plating, pH measurement or HPLC injection.

### 2.5. Statistical Analysis

Statistical analysis was executed using IBM SPSS Statistics v21.0.0.0 (New York, NY, USA). Differences between results which followed a normal distribution (Shapiro–Wilk’s test) were evaluated using one-way ANOVA test coupled with Tukey’s test, with the exception being when comparing between the different times within a given condition. In this case, one-way repeated measures ANOVA test (coupled with Tukey’s test) was used. Differences were considered significant for *p*-values below 0.05.

## 3. Results

As described elsewhere, the extract powder used in the current work mostly comprised anthocyanins (ca. 0.64 mg mL^−1^ of anthocyanins), as it has been subjected to a solid phase extraction (SPE) process that capitalized on acid/neutral fractionation to separate anthocyanins, and their aglycones, from other contaminants like sugars and other carbohydrates that may be present, with most of the non-anthocyanin content of the extract corresponding to anthocyanidins and minerals that originate from the SPE process [1,11,19,20]. Moreover, this extract was selected because it not only resulted from an optimization process that aimed at producing an anthocyanin-rich, food-grade blueberry extract, but also because it had been demonstrated to be effective against an array of potential pathogens infection not only by hindering growth but also adhesion [1,21]. Additionally, in a recent work it was demonstrated that this extract in particular could pose an advantage for a potential probiotic’s colonization while hindering pathogens, which makes the contextualization of interactions much more relevant [11].

Overall, all microorganisms’ growth was affected by the presence of the extract at 1000 μg mL^−1^ (Figure 1), though total OD inhibition throughout the 24 h period was never achieved. From the analysis of Figure 1a, it can be seen that *L. monocytogenes* growth was significantly (*p* < 0.05) reduced in the presence of the extract at 1000 and 500 μg mL^−1^ (57.4% and 19.8% lower than the control, respectively). Furthermore, all concentrations of extract were capable of inducing both a reduction of the maximum OD as well as a reduction of the overall growth rate of *L. monocytogenes*. For *E. coli* (Figure 1b), the extract at 1000 μg mL^−1^ had an OD value that was 52.9% lower than that of the control, at the 24 h mark, while when considering 500 μg mL^−1^ this reduction was of only 36.9%. The remaining two concentrations, while [19] still being capable of significantly (*p* < 0.05) hindering the growth of the bacteria, at the 24 h mark had little to no inhibitory effect. Considering *S. enteritidis* (Figure 1c), it is interesting to note that the extract only inhibited bacterial growth at 1000 μg mL^−1^ (70.3% reduction in OD, compared to that of the control, after 24 h). All other concentrations led to OD values (after 24 h) that were 26.3–40.4% higher than those of the control. A similar behaviour was observed for *B. cereus* (Figure 1d). For this microorganism, the extract was only capable of inhibiting growth at the highest concentration tested (53.1% lower OD than in the control, after 24 h), while the remaining concentrations led to final OD values that were higher than those registered for the control (from 18.1% to 39.6%).

As one of the main focuses of the current work was to characterize the effect of the proposed extract at a concentration capable of inhibiting the pathogenic microorganisms, and only the concentration of 1000 μg mL^−1^, was effective against all four. This concentration was used henceforth. Concentrations above that were not considered as the extract was not soluble at higher concentrations. In Figure 2, the impact of the extract at 1000 μg mL^−1^ (the concentration that appeared to be the most effective in inhibiting bacterial growth (Figure 1)), upon the total viable cells was assessed. As can be seen for both *L. monocytogenes* and *S. enteritidis* (Figure 2a,c), the extract did not allow bacteria to grow as much as they did in the control, as viable cell counts were, on average, ca. 18% and 16% lower, respectively. However, when comparing with the initial bacterial counts, some significant (*p* < 0.05) growth was observed, though it fell below one logarithmic (log) cycle. The same was not observed for *E. coli* (Figure 2b). In this case, there was a significant (*p* < 0.05) reduction in the initial viable cell, which reached 1.82 log of CFU after 12 h. Between 12 h and 24 h, the total viable cells increased by 1.25 log of CFU, though the overall amount still positioned below the one observed in the beginning (0.59 log of CFU lower). For *B. cereus* (Figure 2d) the extract appears to have a bacteriostatic effect, as between 0 and 24 h no significant growth was observed. However, it is interesting to note that, at 12 h, the total viable cell counts had dropped 0.52 log of CFU (*p* < 0.05), meaning that the bacteria counts were being reduced in this time frame.

The extract’s impact upon potential probiotic microorganisms was evaluated and, as can be seen in Figure 3, it had no significant (*p* > 0.05) impact upon the growth of *L. rhamnosus*, *L. plantarum*, *L. acidophilus* and *B.* Bo (Figure 3a–d; bars). The only exception was found for *B*. Bb12. For this microorganism (Figure 3; bars), after 12 h, the presence of extract at 1000 μg mL^−1^ led to a viable count value that was 0.94 log of CFU higher than for the positive control. However, it is important to note that this difference was not observed after 24 h. Furthermore, Figure 3 also displays the acidification of media throughout the assay (Figure 3; lines) and no significant variations were registered between the extract and the control.

The evaluation of the extract’s impact upon the probiotics’ metabolic activity demonstrated that, overall, there was an increase in the amount of acid present (Figure 4), particularly after 24 h. Four different species of acids were identified, lactic, citric and two different short-chain fatty acids (SCFA) viz. acetic and propionic acids. The addition of the extract to the culture media resulted in a ca. 34% reduction in acetic acid concentration at the starting point (Figure 4(a1,a2)). In spite of this, after 24 h the amount of acetic acid found when the bacteria were incubated in the presence of the extract was 1.5–3.16 times higher (for *B.* Bb12 and *B.* Bo, respectively) than that of the control. It is interesting to note that these higher values appeared to be (for all probiotics except *L. rhamnosus*) due to a lack of acetic acid consumption from 12 h onwards because, at that time point, the amount of acetic acid present in the extract is similar or lower than that of the control. As for the effect on citric acid production (Figure 4(b1,b2)) it is interesting to note that, in the case of *L. acidophilus*, the presence of extract appeared to delay the increase in citric acid concentration but, at 12 h and 24 h no statistically significant differences (*p* > 0.05) were found. Similarly, for *B.* Bb12, no significant (*p* > 0.05) differences were found in citric acid concentration after 12 h or 24 h. For all other probiotics, the presence of the extract led to a significant (*p* < 0.05) increase of citric acid levels, ranging from 1.1 to 1.9 times higher than those of the control (for *L. rhamnosus* and *B.* Bo, respectively). Regarding the production of lactic acid (Figure 4(c1,c2)) the extract’s presence led to an increase in the amount of acid produced after a 24 h period (ranging from 1.1 to 3.7 times higher for *L. acidophilus* and *B.* Bo, respectively). The only exception was found for *B.* Bb12, where no statistically significant (*p* > 0.05) differences were found, between extract and positive control, after 24 h. Propionic acid production was also significantly affected by the presence of the extract (Figure 4(d1,d2)). When considering the 24 h mark alone, it can be seen that for all probiotics, there was an increase in propionic acid, ranging from 1.6 to 2.5 times higher for *B.* Bb12 and *L. plantarum*. The only exception was observed for *B.* Bo. In this case, the amount of propionic acid found after 24 h was 1.2 times lower in the presence of the extract. However, the presence of the extract in the media appeared to anticipate the time frame where this acid was produced, e.g., in the positive control, for both *Bifidobacterium*, propionic acid was only observed at 24 h, while, when exposed to the extract, propionic acid was detected after 6 h of incubation.

Figure 5 illustrates the effect of the extract upon probiotic sugar consumption. In regards to glucose consumption (Figure 5(a1,a2)), after 24 h the presence of the extract either had no significant (*p* > 0.05) impact on the leftover glucose (for *L. acidophilus* and *B.* Bb12) or it led to higher values than the control (4.4 and 1.5 times higher for *L. plantarum* and *L. rhamnosus*, respectively). The exception was *B.* Bo for whom the leftover glucose levels were 2.9 times lower than those observed for the control. As for maltose (Figure 5(b1,b2)), and with the exception of *L. rhamnosus*, at the 12 h mark no maltose was detected regardless of the presence of the extract. In the case of *L. plantarum* and *L. acidophilus*, no significant (*p* > 0.05) differences between the extract and control were found at 6 h, hinting at a larger consumption of maltose in the positive control as it had an initial amount of maltose ca. 34% higher than of the media with extract. However, it is interesting to note that, for *B.* Bb12 the opposite appears to be true, i.e., the reduction of maltose concentration at the 6 h mark is significantly (*p* < 0.05) lower in the presence of the extract (70% less in the presence of the extract vs. 25% less in the control). *L. rhamnosus* exhibited a response to the extract presence, in regards to maltose degradation, which was dissimilar to all other microorganisms. More specifically, while in the positive control its fermentative process led to a significant decrease in maltose after 24 h, the amount of maltose found in the presence of the extract after 24 h was 1.2 times higher than that found at the beginning.

## 4. Discussion

The extract, concentrated at 1000 μg mL^−1^, was effective at inhibiting the growth of all food pathogens tested, which stands in accordance to what has been previously reported for an extract, obtained using the same methodology, in regards to other potential pathogens [1]. Additionally, these results are also in line with those reported by Shen et al. [2], who found that *L. monocytogenes* and *S. enteritidis* were susceptible to the action of a blueberry extract. It is interesting to note the disparities between the OD measurements and the quantification of viable cells, as for *L. monocytogenes* and *S. enteritidis* the apparent OD growth did translate into an increase in viable cells, for *B. cereus* and *E. coli*, while the OD hinted at a reduced bacterial growth, the total viable cells either demonstrated no growth (*B. cereus*) or a slight reduction in comparison to the initial bacterial load (*E. coli*). Some differences between both methods have been described early on and may be explained by several reasons, one of which is the accumulation of metabolic products that interfere with the OD measurement or the fact that unviable cells may still be measured by OD, but not in the viable cell determination [22]. Moreover, it is important to note that, for *B. cereus* the work refers only to its effects upon vegetative cells. While an analysis of this was attempted (bacterial cells were submitted to a heat shock 95 °C for 2 min and then plated) no viable cells were detected (data not shown) regardless of the condition, so no conclusions could be made regarding the extract’s effect upon *B. cereus*’s spores.

Anthocyanin-rich extracts have been reported as being effective against both Gram-negative and Gram-positive bacteria by several authors [2,4,6,23,24,25,26]. However, while inhibiting potential pathogens is always interesting, there are bacteria whose inhibition might present a disadvantage, e.g., probiotics. In the present work, none of the probiotic strains’ growth was negatively affected by the extract’s presence, hinting at a selective inhibitory activity. This is similar to what was observed by Lacombe et al. [4] who reported that *Vaccinium angustifolium* blueberry extracts (at 34.75 mg L^−1^ or 17.4 mg L^−1^ equivalents of cyanidin-3-glucoside) were capable of inhibiting the growth of E. coli 157:H7, *L. monocytogenes* and *S. Typhimurium*, while having little to no impact on the growth of *L. rhamnosus*. However, it is interesting to note that, while the blueberry extract proposed in the present work exhibited an antimicrobial activity at significantly higher anthocyanin concentrations (637 mg L^−1^), the higher concentration of anthocyanins had no impact upon probiotic growth. Conversely, it is important to note that the authors did not use an HLPC-based assay (as the one used in this work), but used the differential pH method to quantify anthocyanins, which has been demonstrated to significantly underestimate the anthocyanin values of extracts, so the value considered could be significantly higher than the proposed 34.75 mg L^−1^ or 17.4 mg L^−1^ [4,12]. Puupponen-Pimiä et al. [13] and Puupponen-Pimiä et al. [27] evaluated the effect of both a blueberry extract and a pure anthocyanin (cyanidin-3-glucoside) upon several potential probiotics, among which stand a *L. plantarum* and two *L. rhamnosus* stains, and found that the anthocyanin alone had no inhibitory effect (at concentrations up to 28 μg well^−1^) and that the blueberry extract had no inhibitory effect upon *L. rhamnosus*, therefore standing in line with the lack of inhibitions observed in the present work. Unlike the results proposed in the present work, some inhibitory (growth inhibition but not a reduction of initial viable counts) effects were reported by Biswas et al. [28] when considering the incubation of *Lactobacillus bulgaricus* and *Bifidobacterium bifidum* in the presence of blueberry juice. In a more recent work, using malvidin-3-glucoside (one of the anthocyanins found in blueberries) supplementation in a murine intestinal bowel disease model resulted in a restauration of Firmicutes/Bacteroidetes rations in the diseased animals, which hits that, the current results, while lacking in vivo context are likely to translate into a real-life scenario [29]. Moreso, as most anthocyanins are reported to reach the colon as is, or their conjugates that are transformed with via enterohepatic circulation [30,31].

Cheng et al. [32] described that the presence of an anthocyanin-rich extract exerted a small inhibitory effect upon *L. plantarum* and *L. acidophillus* and a relatively strong inhibition of *B. animalis*. However, these results are contrary to those observed in the current work, not those observed in vivo where anthocyanin and anthocyanin-rich extracts’ supplementation is considered. To the best of our knowledge, no work has focused on the possible metabolic consequence of their presence at levels capable of inhibiting pathogenic growth. Overall, the extracts appeared to cause an increase in the amount of acids produced by the probiotics, at the 24 h mark. Since both *Lactobacillus* and *Bifidobacterium* have been described as being capable of glycosylating anthocyanins (as a likely consequence of β-glucosidase’s activity), in the presence of the extract the amount of sugar monosaccharides present is higher than in the control. In turn, this means that there is a higher amount of sugars to be used in fermentative processes, therefore the amount of lactic acid produced would be higher and thus, there is more to be forwarded down the metabolic pathways into SCFA biosynthesis [33,34,35,36]. Moreover, these results stand in line with those described by Mousavi et al. [37] and Zhang et al. [38], who reported that the fermentation of an anthocyanin-rich pomegranate juice and *Opuntia ficus-indica* resulted in higher levels of both SCFA. Organic acids, particularly SCFA, are probiotic metabolites that have been widely associated with an array of health-promoting properties [39,40]. Namely, *L. plantarum* 299v has been described as producing propionic and acetic acids, SCFA that have been associated with the inhibition of pathogenic microorganisms [41]. Overall, this might mean that the antimicrobial activity of the extract could be accentuated by the presence of the probiotic bacteria, as all acid levels were significantly higher when probiotics were exposed to the extract, and may thus contribute to their inhibition. However, not all acid increases may be advantageous from a health standpoint as the increase in acetic acid absorption has been linked with an increase in serum cholesterol levels while propionic acid has been reported as an inhibitor of cholesterol biosynthesis [42,43,44]. As the proposed extract leads to both acetic acid accumulation and an earlier production of propionic acid, to speculate on its potential impact on cholesterol levels is precocious. From a different perspective, the conservation of these acids after 24 h may be interesting for fermented products as it may allow for the extension of product shelf life as the acids may act not only as antimicrobials, but also as antioxidants and texture/colour stabilizers [45,46,47,48,49].

From a sugar-consumption standpoint, it can be seen that the presence of the extract caused a reduction in the consumption of glucose. Considering that higher amounts of acids were produced from a smaller amount of sugars, it stands to reason that some other compounds are acting as a substrate for the potential probiotics, and as anthocyanins have been demonstrated to act as a possible carbon source, hypothetically they may also be acting as a substrate in this case [32].

It is interesting to highlight that the addition of the extract to the culture media affected both the amount of acetic acid and maltose present at the beginning of the incubation. While interactions between the extract and culture media are out of the scope of the present work, the existing literature provides some insights into why this may be observed. Anthocyanin’s acetylation is a relatively well-described process and it may explain the reduction of acetic acid as it would be sequestered by the anthocyanin molecules [50,51]. Maltose reduction, however, may not be as easy to explain, though other authors have also found that the addition of this sugar to anthocyanin-rich extracts causes a small reduction in the total anthocyanins [37,52]. However, Jackman et al. [53] reported that some sugars, and their degradation products (resulting from Maillard reactions and other oxidation reactions), may cause a reduction in the detected anthocyanin values.

Overall, it is important to bear in mind that these results lack specific biological contexts that may play an important role when considering real-life applications. This is particularly critical when considering the potential effect upon the gut microbiota context. Of relevance stands the interaction with other gut microbiota bacteria, and the way the exposure is made [54,55]. While *Lactobacillus* and *Bifidobacterium* are reported as being the major groups responsible for glycosylating anthocyanins, several other members of the microbiota play a role in the subsequent degradation of the aglycones. Moreover, these metabolites (which are not contemplated in the current work) will also have an important role in the different biological effects among which stands their antimicrobial effect [8,56,57]. Also, in the real-life scenario bacteria are not always in a planktonic state as those considered in the current work, they are frequently found in established communities. This is particularly true in the intestine where bacteria are present not only in the digested matter, but also in its wall and the mucous that surrounds it [58,59].

## 5. Conclusions

The hereby proposed extract, at 1000 μg mL^−1^, was capable of effectively inhibiting the growth of the four potential pathogenic strains considered in this work with *E. coli* appearing to be the most susceptible to its presence, followed by *B. cereus* and less so *L. monocytogenes* and *S. enteritidis.* However, while effectively inhibiting the growth of four potential pathogenic microorganisms, the same concentration of extract did not hamper probiotics growth, with the extract’s presence exerting little impact upon the viable probiotic counts. In fact, the extract’s effect was only observed at a metabolic level, as its presence resulted in, overall, higher amounts of organic acids’ accumulation in the media (when in comparison to the control). While the accumulation of organic acids may be interesting from a food-production standpoint, the accumulation of acetic acid may not be as interesting from a health-promotion standpoint (as acetate functions as a precursor for cholesterol synthesis). However, this accumulation of acetic acid is also accompanied by an increase in the production of propionic acid, which has some interesting health promoting potential, thus demonstrating the need of further studies in order to better elucidate the blueberry extract potential effects. In sum, it can be concluded that the hereby proposed extract poses an interesting solution when seeking to control potential pathogens while exerting a positive effect upon potential probiotics, and thus exhibit more than one potentially beneficial effect.

## Figures and Tables

**Figure 1 foods-12-00734-f001:**
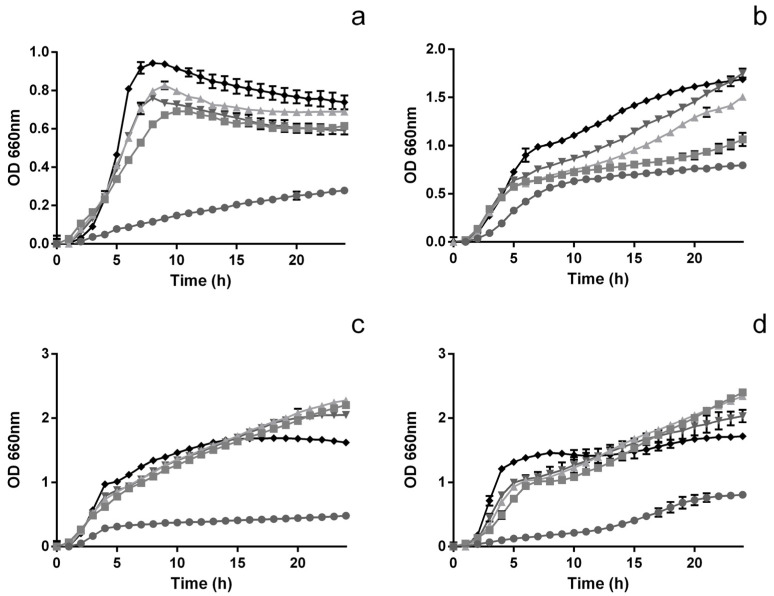
Time inhibition curves for *L. monocytogenes* (**a**), *E. coli* (**b**), *S. enteritidis* (**c**) and *B. cereus* (**d**) when exposed to different concentrations of extract; 1000 μg mL^−1^ (●), 500 μg mL^−1^ (■), 250 μg mL^−1^ (▲), 125 μg mL^−1^ (▼) and 0 μg mL^−1^ (◆).

**Figure 2 foods-12-00734-f002:**
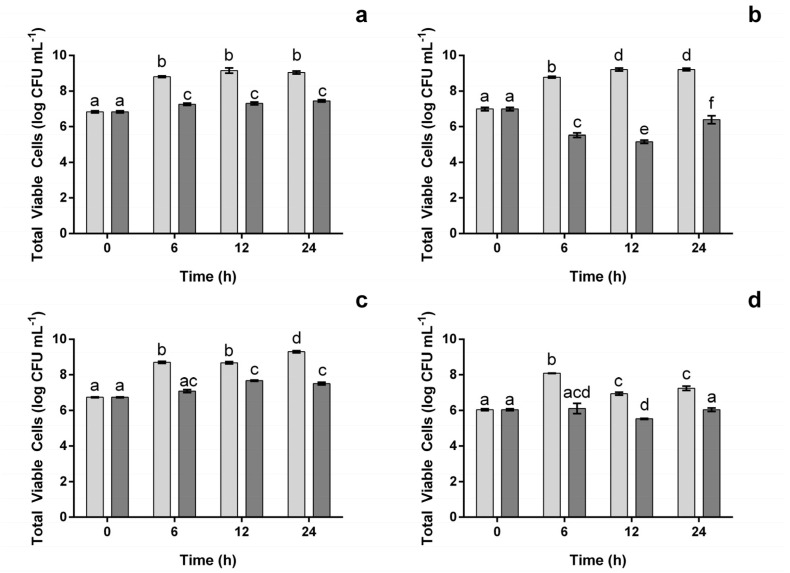
Total viable cells for *L. monocytogenes* (**a**), *E. coli* (**b**), *S. enteritidis* (**c**) and *B. cereus* (**d**) in the presence ((■) 1000 μg mL^−1^) and absence (■) of the extract. The different letters represent the statistically significant (*p* < 0.05) differences between each bar.

**Figure 3 foods-12-00734-f003:**
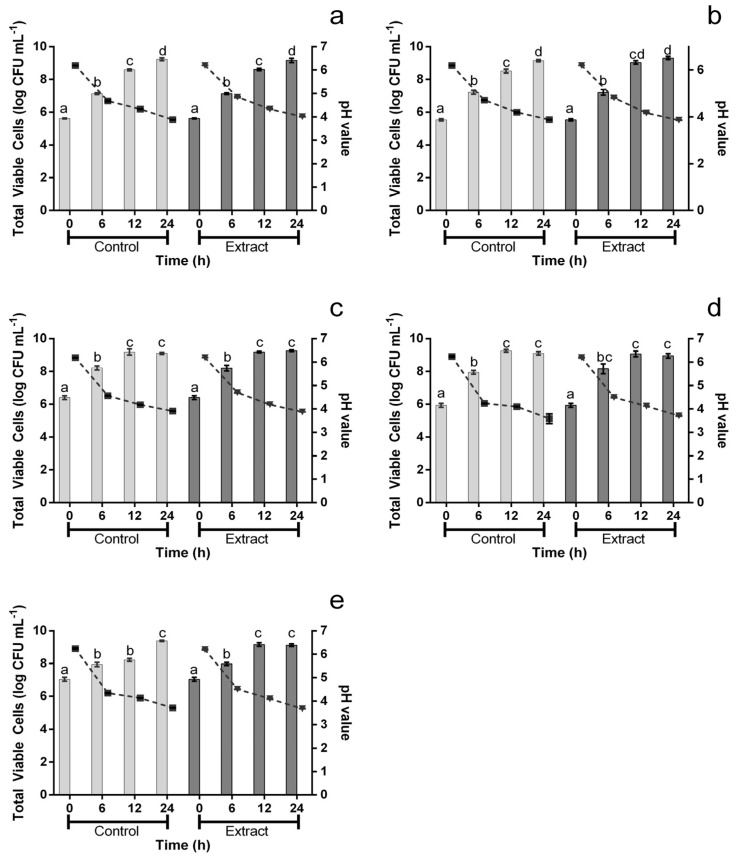
Total viable cells (bars) and pH values (lines) for *L. rhmamnosus* (**a**), *L. plantarum* (**b**), *L. acidophilus* (**c**), *B*. Bo (**d**) and *B*. Bb12 (**e**) when exposed, or not, to 1000 μg mL^−1^ of the extract. Different letters mark the statistically significant (*p* < 0.05) differences between the bars.

**Figure 4 foods-12-00734-f004:**
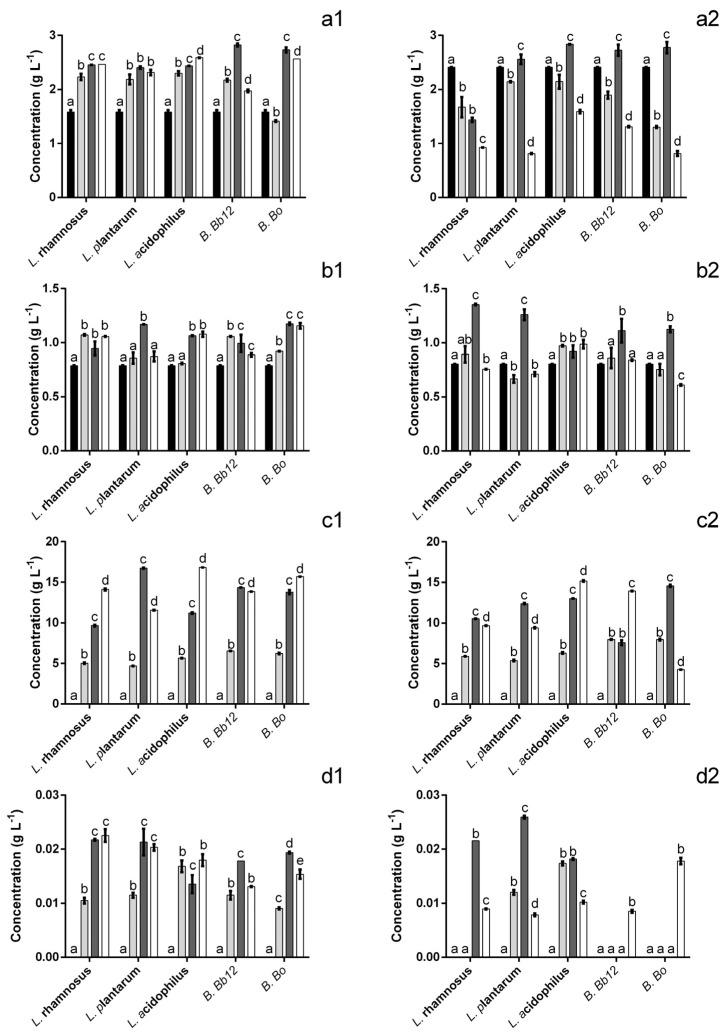
Concentration of acetic (**a**), citric (**b**), lactic (**c**) and propionic (**d**) acids in the presence (1) and absence (2) of extract at 0 (■), 6 (■), 12 (■) and 24 (☐) h. The different letters mark the statistically significant (*p* < 0.05) differences between the times for each individual microorganism assayed.

**Figure 5 foods-12-00734-f005:**
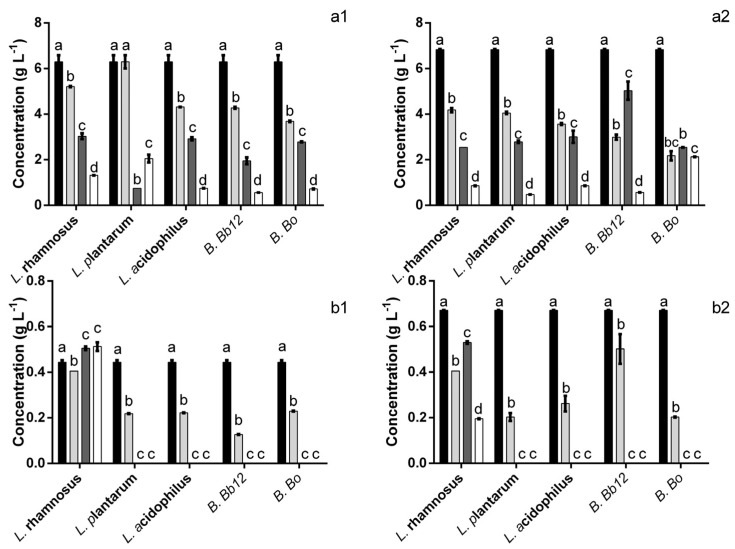
Concentration of glucose (**a**) and maltose (**b**) in the presence (1) and absence (2) of extract at 0 (■), 6 (■), 12 (■) and 24 (☐) h. The different letters mark the statistically significant (*p* < 0.05) differences between the times for each individual microorganism assayed.

## Data Availability

The data presented in this study are available on request from the corresponding author. The data are not publicly available due to confidentiality agreements.

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
