# Peer review of "Selective Activity of an Anthocyanin-Rich, Purified Blueberry Extract upon Pathogenic and Probiotic Bacteria"

_foods, 2023, doi:10.3390/foods12040734_

Round 1

Reviewer 1 Report

Blueberry extracts have been widely recognized as possessing antimicrobial activity against several potential pathogens. However, the contextualization of the interaction of these extracts with beneficial bacteria (i.e. probiotics), particularly when considering food applications of these products, may be of importance, not only because their presence is important in the regular gut microbiota but also because they are important constituents of regular and functional foodstuffs. Results showed that the extract, at a concentration that inhibited L. monocytogenes, B. cereus, E. coli and S. enteritidis (1000 µg mL-1), had no inhibitory effect on the growth of the potential probiotic stains used. However, the results demonstrated, for the first time, that the extract had a significant impact on metabolic activity of all probiotic strains resulting in higher amounts of organic acid production and an earlier production of propionic acid.

It is well design and the result is interesting. The manuscript has been revised and the quality has been improved. However, there still have minor revision need to improve.

1. The sentence should be checked in the manuscript.

2. The polyphenols has potential effect of prebiotic effect and inhibit harmful bacteria, even regulate gut microbiota (Oat phenolic compounds regulate metabolic syndrome in high fat diet-fed mice via gut microbiota. Food Bioscience. 50(2022)101946. Doi: 10.1016/j.fbio.2022.101946.).

3. The phenolic acid content should be measure in order to get the positive correlation.

4. “2.1. Extract Production and Purification”

It is better to measure the pectin content, which may affect the prebiotic effect (Whole grain benefit: oat β-glucan and phenolic compounds synergistically regulates hyperlipidemia via gut microbiota in high-fat-diet mice. Food & Function, 2022, 13, 12686-12696. Doi: 10.1039/d2fo01746f.).

The reference should be update in recently years.

Author Response

Blueberry extracts have been widely recognized as possessing antimicrobial activity against several potential pathogens. However, the contextualization of the interaction of these extracts with beneficial bacteria (i.e. probiotics), particularly when considering food applications of these products, may be of importance, not only because their presence is important in the regular gut microbiota but also because they are important constituents of regular and functional foodstuffs. Results showed that the extract, at a concentration that inhibited L. monocytogenes, B. cereus, E. coli and S. enteritidis (1000 µg mL-1), had no inhibitory effect on the growth of the potential probiotic stains used. However, the results demonstrated, for the first time, that the extract had a significant impact on metabolic activity of all probiotic strains resulting in higher amounts of organic acid production and an earlier production of propionic acid.

It is well design and the result is interesting. The manuscript has been revised and the quality has been improved. However, there still have minor revision need to improve.

Q1. The sentence should be checked in the manuscript.

Reply Q1: We appreciate the reviewers’ input but we failed to understand exactly which sentence the reviewer would like for us to check.

Q2. The polyphenols has potential effect of prebiotic effect and inhibit harmful bacteria, even regulate gut microbiota (Oat phenolic compounds regulate metabolic syndrome in high fat diet-fed mice via gut microbiota. Food Bioscience. 50(2022)101946. Doi: 10.1016/j.fbio.2022.101946.).

Reply Q2: We understand your point of view and have altered the manuscript to better portray it, including the reference provided (lines 43 – 46).

Q3. The phenolic acid content should be measure in order to get the positive correlation.

Reply Q3: Thank you for your input, we are unsure if we understood the question correctly, but the sample has been purified and contains 0.637 mg/mL of anthocyanins, with the remainder being, mostly salt that results from the purification process itself. We have have strived to make this clearer in the manuscript (lines 119 – 125).

Q4. “2.1. Extract Production and Purification”

It is better to measure the pectin content, which may affect the prebiotic effect (Whole grain benefit: oat β-glucan and phenolic compounds synergistically regulates hyperlipidemia via gut microbiota in high-fat-diet mice. Food & Function, 2022, 13, 12686-12696. Doi: 10.1039/d2fo0174          6f.).

Reply Q4: While we appreciate the reviewers’ opinion, the extract produced was extracted using ethanol (which is usually used to precipitate pectins, and not extract them) (1). Moreover, they were then further purified using Bond Elut Plexa SPE columns which are suitable to separate compounds using acid/neutral fractionation, which is suitable for anthocyanins, and allowed for the removal of other interferents like sugars and other carbohydrates and even most other phenolic compounds that were not anthocyanins and their aglycones (as can be seen in references 2,3 which refer to two different extract production batches) (2,3,4).

References:

  1. Koh J, Xu Z, Wicker L. 2018. Blueberry pectin extraction methods influence physico-chemical properties. J Food Sci 83(12):2954.
  2. Silva S, Costa EM, Oliveira H, De Freitas V, Morais RM, Calhau C, Pintado M. 2022. Impact of a purified blueberry extract on in vitro prebiotic mucin-adhesion and its effect on probiotic/intestinal pathogen systems. Molecules 27(20):6991.
  3. Silva S, Costa EM, Mendes M, Morais RM, Calhau C; Pintado MM. 2016. Antimicrobial, antiadhesive and antibiofilm activity of an ethanolic, anthocyanin-rich, blueberry extract purifies by solid phase extraction. J Appl Microbiol 121:693.
  4. Sandhu A, Edirisinghe I, Burton-Freeman B, Zweigenbaum J. UHPLC-MS/MS Triple analysis of anthocyanin metabolites in Human plasma using protein precipitation and solid phase extraction for determination of uptake from food – Aplication note from Agilent (https://hpst.cz/sites/default/files/oldfiles/5991-6526en.pdf )

Reviewer 2 Report

The manuscript entitled "Selective activity of an anthocyanin-rich, purified blueberry extract upon pathogenic and probiotic bacteria" highlights the active compounds of blueberry on bacterial growth, particularly on pathogenic and beneficial bacteria. With the increase of AMR microbes in the environment and public health, alternatives to antimicrobials having effectiveness against pathogenic microbes are needed at the time. 

This research presents the effectiveness of blueberry extracts against certain types of pathogenic bacteria with no or little effect on beneficial bacteria. Although the manuscript is written well with appropriate methodology and results showing significance of the result, there are some questions which need comments or discussion.

1. I think there is the problem with the adjustment or formatting of figures in the manuscript or maybe disturbed during conversion to pdf format. However, the figure needs to be adjusted for a good presentation. 

2. The results show the production of acetic acid, citric acid, lactic acid, and propionic acid in higher concentrations in response to blueberry extract subjected to in-vitro bacterial cultures. The in-vitro effect may vary from in-vivo effect owing to various factors like bacterial biofilms. I suggest authors to highlights or at least discuss the limitations or shortcomings of this research in manuscript.

3. Furthermore, the results showing increase in lactic acid concentrations but the authors didnt discussed the effects produced by lactic acid on bacterial growth etc as they have discussed for acetic and propionic acids. 

4. Conclusion section showing deficienies and should be improved to support the results. The conclusion is more focussed on organic acid production rather than the antimicrobial properties of blueberry extract. 

Author Response

The manuscript entitled "Selective activity of an anthocyanin-rich, purified blueberry extract upon pathogenic and probiotic bacteria" highlights the active compounds of blueberry on bacterial growth, particularly on pathogenic and beneficial bacteria. With the increase of AMR microbes in the environment and public health, alternatives to antimicrobials having effectiveness against pathogenic microbes are needed at the time. 

This research presents the effectiveness of blueberry extracts against certain types of pathogenic bacteria with no or little effect on beneficial bacteria. Although the manuscript is written well with appropriate methodology and results showing significance of the result, there are some questions which need comments or discussion.

Q1. I think there is the problem with the adjustment or formatting of figures in the manuscript or maybe disturbed during conversion to pdf format. However, the figure needs to be adjusted for a good presentation. 

Reply Q1: Thank you for your attention, the problems did arise from the pdf conversion we have attempted to resolve by replacing the figures.

Q2. The results show the production of acetic acid, citric acid, lactic acid, and propionic acid in higher concentrations in response to blueberry extract subjected to in-vitro bacterial cultures. The in-vitro effect may vary from in-vivo effect owing to various factors like bacterial biofilms. I suggest authors to highlights or at least discuss the limitations or shortcomings of this research in manuscript.

Reply Q2: We appreciate the input. We understand your point and have strived to better illustrate this in the discussion (lines 332-344).

Q3. Furthermore, the results showing increase in lactic acid concentrations but the authors didnt discussed the effects produced by lactic acid on bacterial growth etc as they have discussed for acetic and propionic acids. 

Reply Q3: We appreciate the reviewer’s point of view and have altered the discussion section to better incorporate the required information (lines 290-296; 303-304).

4. Conclusion section showing deficienies and should be improved to support the results. The conclusion is more focussed on organic acid production rather than the antimicrobial properties of blueberry extract.

Reply Q4: We understand the reviewers point of view and have altered the conclusion accordingly (lines 346-364)

Reviewer 3 Report

The manuscript provides new insights regarding the use of blueberry extracts on pathogenic and BAL. Beside the effects on pathogenic microorganism be recognized, authors provided new information on its effects on BAL, including metabolism. The manuscript is very clear and well-written, and I just have few remarks before recommending its publication.

Major remarks

-        Authors must discuss their results of B. cereus considering the vegetative cells and spores. There is no mention to spores. Were OD and counts different due spores survive? The discussion section must discuss the different effects on both type of cells.

-        Authors mentioned “triplicate” for the microbiological analyses. However, was the study done once? No replicates were done?

-        Conclusion section sounds as a discussion. Authors must proper conclude based on their results.

Minor remarks

-        Authors must standardize the use of abbreviations of scientific names. I suggest only abbreviating the genus and use the full specie.

-        Typhymurium is a serovar and must be correctly written.

-        The citations in text also need to be standardized. Sometimes appears five authors names instead of using “et al” or “colleagues”. This must turn easier for readers.

Author Response

The manuscript provides new insights regarding the use of blueberry extracts on pathogenic and BAL. Beside the effects on pathogenic microorganism be recognized, authors provided new information on its effects on BAL, including metabolism. The manuscript is very clear and well-written, and I just have few remarks before recommending its publication.

Major remarks

-        Q1:Authors must discuss their results of B. cereus considering the vegetative cells and spores. There is no mention to spores. Were OD and counts different due spores survive? The discussion section must discuss the different effects on both type of cells.

Reply Q1: Thank you for your question. When plating B. cereus samples, we did a scan for the effect upon spore formation (we gave the samples a heat shock of 95ºC, 2min, and then diluted following the procedure described in the manuscript) but since neither the control nor the samples exhibited any growth, we opted to not mention data regarding spores because the conditions used did not appear to result in spore formation. We have altered the discussion section to include this (lines 254-258).

-        Q2: Authors mentioned “triplicate” for the microbiological analyses. However, was the study done once? No replicates were done?

Reply Q2: We only mentioned the independent assays and not the replicates within each experiment. We have altered the materials and methods section to better illustrate the replicates used (lines 81-82; 88-89; 105-106).

-        Q3: Conclusion section sounds as a discussion. Authors must proper conclude based on their results.

Reply Q3: We appreciate the reviewers input and have altered the conclusion’s section (lines 346-364)

Minor remarks

-       Q4:  Authors must standardize the use of abbreviations of scientific names. I suggest only abbreviating the genus and use the full specie.

Reply Q4: While we appreciate the reviewer’s point of view and have checked to ensure that all references to bacterial names are written by abbreviating as proposed. An exception was made for Bifidobcterium animalis since we use two different strains of the same species. In this particular case we abbreviated using the genus and the strain reference to avoid confusion.

-        Q5: Typhymurium is a serovar and must be correctly written.

Reply Q5: We have altered the manuscript. Thank you for your attention. (line 268)

-        Q6: The citations in text also need to be standardized. Sometimes appears five authors names instead of using “et al” or “colleagues”. This must turn easier for readers.

Reply Q6: We appreciate the attention provided, and we have altered the manuscript accordingly.

Round 2

Reviewer 1 Report

Blueberry extracts have been widely recognized as possessing antimicrobial activity against several potential pathogens. However, the contextualization of the interaction of these extracts with beneficial bacteria (i.e. probiotics), particularly when considering food applications of these products, may be of importance, not only because their presence is important in the regular gut microbiota but also because they are important constituents of regular and functional foodstuffs. Results showed that the extract, at a concentration that inhibited L. monocytogenes, B. cereus, E. coli and S. enteritidis (1000 µg mL-1), had no inhibitory effect on the growth of the potential probiotic stains used. However, the results demonstrated, for the first time, that the extract had a significant impact on metabolic activity of all probiotic strains resulting in higher amounts of organic acid production and an earlier production of propionic acid.

It is well design and the result is interesting. The manuscript has been revised and the quality has been improved. However, there still have issues need to improve.

1. Since the author design to study the correlation about pathogenic and probiotic bacteria. This paper should be referred and cited (The positive correlation of antioxidant activity and prebiotic effect about oat phenolic compounds. Food Chemistry, 402(2023): 134231.).

2. Does the blueberry extracts exist other polysaccharide or pectin that can effect the potential probiotic? (Whole grain benefit: oat β-glucan and phenolic compounds synergistically regulates hyperlipidemia via gut microbiota in high-fat-diet mice. Food & Function, 2022, 13(24), 12686-12696. Doi: 10.1039/d2fo01746f.)

3. “2.1. Extract Production and Purification”

“further purified using Bond Elut Plexa SPE columns which are suitable to separate compounds using acid/neutral fractionation” should refer this paper (Quantitative determination of Nepsilon-(carboxymethyl)lysine in sterilized milk by isotope dilution UPLC-MS/MS method without derivatization and ion pair reagents. Food chemistry. 385(2022): 132697.)

4. There should be significant different comparison when treatment with different concentrations of extract.

5. Statement expression needs to be improved.

6. References should be updated, especially the refence that were not in recent years.

Author Response

Blueberry extracts have been widely recognized as possessing antimicrobial activity against several potential pathogens. However, the contextualization of the interaction of these extracts with beneficial bacteria (i.e. probiotics), particularly when considering food applications of these products, may be of importance, not only because their presence is important in the regular gut microbiota but also because they are important constituents of regular and functional foodstuffs. Results showed that the extract, at a concentration that inhibited L. monocytogenes, B. cereus, E. coli and S. enteritidis (1000 µg mL-1), had no inhibitory effect on the growth of the potential probiotic stains used. However, the results demonstrated, for the first time, that the extract had a significant impact on metabolic activity of all probiotic strains resulting in higher amounts of organic acid production and an earlier production of propionic acid.

It is well design and the result is interesting. The manuscript has been revised and the quality has been improved. However, there still have issues need to improve.

  1. Since the author design to study the correlation about pathogenic and probiotic bacteria. This paper should be referred and cited (The positive correlation of antioxidant activity and prebiotic effect about oat phenolic compounds. Food Chemistry, 402(2023): 134231.).

Reply Q1: The reference has been added to the manuscript.

  1. Does the blueberry extracts exist other polysaccharide or pectin that can affect the potential probiotic? (Whole grain benefit: oat β-glucan and phenolic compounds synergistically regulates hyperlipidemia via gut microbiota in high-fat-diet mice. Food & Function, 2022, 13(24), 12686-12696. Doi: 10.1039/d2fo01746f.)

Reply Q2: The extract contains, no other polysaccharide but the reference has been added to the manuscript.

  1. “2.1. Extract Production and Purification”

“further purified using Bond Elut Plexa SPE columns which are suitable to separate compounds using acid/neutral fractionation” should refer this paper (Quantitative determination of Nepsilon-(carboxymethyl)lysine in sterilized milk by isotope dilution UPLC-MS/MS method without derivatization and ion pair reagents. Food chemistry. 385(2022): 132697.)

Reply Q3: The reference has been added to the manuscript.

  1. There should be significant different comparison when treatment with different concentrations of extract.

Reply Q4: We appreciate the reviewers point of view. There are statistically significant differences in pathogen behaviour when exposed to different concentrations of extract as can be seen in Figure 1. However, as the aim was to assess the effect of a concentration capable of inhibiting pathogenic bacteria, in the remaining assays only one concentration of extract was tested (higher concentrations were not possible due to the extract’s solubility. We have altered the manuscript to make this clearer.

  1. Statement expression needs to be improved.

Reply Q5. We appreciate the reviewer’s input. We have strived to improve the clarity.

  1. References should be updated, especially the refence that were not in recent years.

Reply Q6. We altered the references as requested.